# Evaluating the Sustainability Dimensions in the Food Supply Chain: Literature Review and Research Routes

Maria Elena Latino *, Marta Menegoli and Martina De Giovanni

Innovation Engineering Department, University of Salento, 73100 Lecce, Italy;
marta.menegoli@unisalento.it (M.M.); martina.degiovanni@unisalento.it (M.D.G.)
* Correspondence: mariaelena.latino@unisalento.it; Tel.: +39-0832-297949

**Abstract:** Nowadays, the world is facing numerous sustainability challenges and the modern food system is called to innovate processes or products in order to remain competitive within the market, as well as answering to strategic government guidelines for a more sustainable food supply chain. This study aims to investigate what the main research routes of a sustainable food supply chain are, explored by the international scientific panorama, with a view for providing companies with a framework of the sustainability paths that can be followed, and, to researchers, gaps and future research routes to explore. A systematic review method is adopted through bibliometric analysis and results were obtained with VOSViewer software support. Descriptive and thematic analyses allowed us to discover the bibliometric characteristics of the sample, the main specific topics and the related research routes already addressed in sustainable food supply chain, the main food supply chain models studied in association with sustainability and the effort employed by academia to investigate the three sustainability dimensions: environmental, economic and social. Concluding, the research field of sustainability in the food supply chain is focused on management issues able to generate impacts on process, systems, practices, production and quality.

**Keywords:** sustainability; food supply chain; economic sustainability; social sustainability; environmental sustainability; systematic review; bibliometric analysis

## 1. Introduction

Following the neoclassical perspective, the food system is a socio-economic structure where (i) individuals' behavior is guided by rational choice, decision making, human capital factors and lifestyle choices [1] and (ii) different types of sub-systems co-exist and reflect different ways of producing, processing, distributing and consuming food products [2]. The current food system is recognized as dynamic and complex, like other systems for example in manufacturing context [3], with a large number of actors capable of performing many activities around the word and according to several production models, bringing the food "from farm to fork". Globalization has impacted the food system, characterizing it by intensification, specialization, international sourcing, concentration and homogenization of food production and consumption [4]. In order to consider all these features, several conceptual approaches to the food system have been developed over the years within international scientific panorama, such as: system of provision [5], product lifecycle [6], industrial ecosystem [7,8] and food supply chain [9,10]. Specifically, food supply chain is a system of phases or stages, which represents a sequence of activities through which resources, materials and information flows are facilitated both downstream and upstream, in order to produce goods and/or services for consumption or utilization by a consumer [4,11]. The food supply chain is seen as a network of organizations and multiple actors which, through mutual contracts and economic relations, enable all the steps needed to produce and move foods from field to fork (agricultural production, storage and distribution, processing and packaging, retail and marketing). Farmers, processors, wholesalers, transporters and retailers are some of the actors involved in food supply

chains [12]. The steps in the food supply chain are all connected and changes to one step affect the others along the chain, highlighting the potential of impacting on one or more sub-systems of food system: supply chain activities—such as processing—affect a food product's nutritional quality and affordability. Food supply chains operate at different scales and levels. The number of economic operators, the breadth of economic development and geographical and social relations between food supply chain actors determine the "length" of the food supply chain. Therefore, the long supply chain is characterized by many phases and actors, a global economic development, a broad geographical coverage and extensive social relationships. Conversely, limited phases and actors, a local economic development, a local geographical coverage and confined social relations characterize the short food supply chain. In rural and isolated communities, food supply chains may be short: farmers and food producers either eat the food directly or sell it in the local market. In large urban settings, food supply chains may be longer and more complex: food is typically produced farther away and more people are involved in its production, processing, packaging and retail [13]. These diversities are also recognizable considering the supply chain of specific food product categories; for instance, local or perishable foods are generally produced along a short supply chain, while imported/exported food requires a long supply chain. Thanks to the continuous social evolution and technological progress that we can see over time around the world, food supply chains are undergoing rapid transformations, especially in low and middle-income countries, often leading to more interaction between different chains and enlarging their impact range [14]. All food supply chains impact on the environment, economy and society of the territories on which they settle [15], giving to researchers and practitioners the possibility and the interest to assess and study how make it more sustainable in order to achieve strategic resilience [16].

### 1.1. Sustainability Challenges in the FSC

Global sustainability issues increase the complexity of the modern food supply chains. World hunger [17], global population growth and suspected future food unavailability [18], climate change [19], agro-biodiversity safeguard [20], rural area protection [21], pandemic condition [22], food safety [23], trust in food supply chains [24], food waste reduction [25], environmental impact and alternative production methods recognition capable of reducing it [26] and pesticide utilization in farming phase [27] are some of the biggest challenges that the modern food system is facing. Technological improvements, widespread use of agricultural chemicals, modern farm machinery and advanced transportation systems lead food supply chains to produce and supply surpluses of food [28]; however, this seems to be not sufficient in order to solve, totally, the problems and, in some cases, increased the growth of environmental and social concerns.

In this context, sustainability is understood according the triple-bottom line approach [29] that, balancing the three sustainability dimensions of economy, environment and society, is capable of fostering sustainable development: "development that meets the needs of the present without compromising the ability of future generations to meet their own needs" [30]. Governments over the world are building strategic frameworks in order to sensitize, support and guide food supply chains along sustainable innovation processes [31,32]. The European Union (EU), for example, legally defines short supply chain within the rural development regulation (1305/2013) and considers it an enabler of "sustainable agriculture" because it is able to reach environmental sustainability goals through the reduction of transportation costs and consequently of $CO_2$ emissions [33]. In addition, with the Reg. 1305/13, the EU promotes biodiversity and implements peri-urban agriculture. Thus, the interest for short food supply chain is growing in the EU and in national legislations, considering its role in achieving environmental goals. At the same time, the academy has also extensively discussed the adoption of sustainable practices in food supply chains, analyzing the current challenges [34] and proposing production models [35], technologies [36] or business models [37]. From the scientific panorama, it arose that making food supply sustainable should be possible through several initiatives [38]:

using resource efficiency [39]; improving management processes [40]; raising visibility and awareness about sustainability issues among all stakeholders [41]; increasing collaborative relationships with suppliers and customers [42]; implementing food traceability [43]; shortening the chain [44]; assessing and reducing all greenhouse gas emissions related to the business processes [45]; protecting rural areas [46]; respecting labor rights [47]; controlling biological crop or livestock status for pest management [48]. Initiatives need a clear and coherent development strategy: the main objectives should be clear and transparent to every actor in the supply chain [38]. Only when the strategy is supported by all the actors along the chain will the initiative be successful and the food supply chain become a sustainable food supply chain.

### 1.2. Research Gap and Purpose Statement

Some efforts have also been made to systematize the literature about the sustainable food supply chain. In a recent literature review [44], the state of the art of the definition and characterization of the food supply chains, especially the short ones, and their sustainability was provided. Further, [49] addressed the issues of sustainable food supply chain from the operations research viewpoints, focusing on the identification of the mathematical modelling techniques adopted by researchers in the analytical modelling of sustainable food supply chain. As well, [50] reviewed the sustainable supply chain management subject with the aim of describing the practices that allow companies to maintain control over the supply chain. Finally, [51] proposed a review about quantitative models for sustainable food logistics management with the aim to discover the key logistic scopes. To the best of our knowledge, no study is concerned with investigating what the main research routes of sustainable food supply chain explored by the international scientific panorama are, providing to food companies a framework of the sustainable paths in place along the supply chain and to researchers gaps and future research routes to explore. With the aim of filling this emerging gap, this paper proposes a systematic literature review conducted according the PRISMA guidelines. Descriptive and thematic analyses were performed to discover (i) the bibliometric characteristics of the sample, (ii) the main specific topics and the related research routes addressed by the current studies in sustainable food supply chain and (iii) the main food supply chain models studied in association with sustainability. Specifically, these models could be logistic models or models able to describe the specific FSC features for a product or a family of products [52], and (iv) the effort employed to investigate the three sustainability dimensions (environmental, economic and social). These represent the original contribution of the study to the current literature on sustainable food supply chains.

The structure of the paper shows six sections: the first aims to introduce the research background and define the purpose statement of the study; the second section explains the methodology that the analysis leverages on; the third shows results of the study, both descriptive and thematic; the fourth encompasses discussions based on the results of the study; the fifth reports some conclusion marks, implications and limitations of the study.

## 2. Methodology

### 2.1. Choosing a Review Methodology

The systematic review method is helpful in outlining the boundaries of knowledge [53] and was applied to identify and critically analyse contributions to the research topic [54].Using this method, a researcher is guided in the systematization and sharing of the results about a specific body of literature [55,56]. Moreover, this methodology is useful when the purpose of the study is to map a research field, identify research gaps, and develop an agenda for further research [57]. Leveraging these bases, the researchers chose to adopt a systematic review methodology in the present study. This method allows for the selection, through an interrogation of databases by a well-structured query based on ex-ante keywords definition and of reference literature that needs to be analysed to create evidences. Specifically, among the several kinds of systematic literature reviews proposed by

Paul and Criado [58,59], this study adopts the domain-based review. Following PRISMA guidelines, the proposed systematic literature review considers three sequential steps: (1) questions formulation; (2) definition of the protocol for review; (3) analysis of the results in terms of descriptive and thematic analysis as well as data synthesis. The review process was concluded by proposing descriptive analysis of the sample and the thematic map of the knowledge that surrounds the research topic: sustainable food supply chain.

### 2.2. Questions Formulation

According to Cronin [60], the systematic literature review method must start with the definition of a focused research question, able to guide the review process. Starting from this research question, search strings for the scientific database searches are defined. According to the research premise discussed above, the research questions were specified as follows:

*"What are the main research routes addressed by the studies on sustainable food supply chain?".*

*"What are the main food supply chain models studied in association with sustainability?"*

*"Is one of the sustainability dimensions more investigated than the other ones?"*

### 2.3. Definition of the Protocol for Review

To identify the sample of analysis, a search scheme and the inclusion and exclusion criteria needed to be defined. Search keywords were used to identify and carefully select the relevant literature by defining a well-structured query, combining representative keywords through Boolean operators [61]. Considering the research topic and using the AND Boolean operator, the following search scheme was defined: ("food supply chain" AND "sustainability"). The identified query was used for questioning one important indexed electronic scientific database, in order to identify the relevant studies in the scientific literature. The selected database was Scopus (http://www.scopus.com, accessed on 3 September 2021), managed by Elsevier publishing. It is considered to be one of the most extensive databases [62] because it is more comprehensive than other databases, such as Web of Science, which provides only ISI indexed journals [63]. Taking into account these evidences and according to the choice made in previous studies [64,65], the selection of the Scopus scientific database assures the research validity of the study.

The application of the search scheme in the Scopus data base and the consequential identification of the initial sample were performed on 3 September 2021; thus, this study considered papers published until this date. The research was conducted in the 'Article title, Abstract, Keywords' field of Scopus setting, in order to collect contributions on theme. No temporal restrictions were imposed. The selected keywords provided an initial sample of 736 papers. With the aim to export a homogeneous data set, these papers were selected by applying one inclusion and one exclusion criteria: the inclusion criterion was that the study was written in English; the exclusion criterion was that the study is an editorial, a data paper, an erratum or a note. Consequently, a final sample of 716 papers was recognised. Export in csv and BibTex file allowed the researches to start the data analysis phase and facilitate the citation process.

### 2.4. Analysis of the Results and Data Synthesis

This study performs two complementary analyses of the bibliometric analysis: performance analysis and Science Mapping. Performance analysis seeks to evaluate the research and publication performance of individuals and institutions [66]. It was based on bibliographic information of the sample and was conducted in this study, by counting and statistical methods, using Excel software. It allows for providing an overview of the sample characteristics. Science Mapping aims to discover the structure and dynamics of a scientific field, focusing on the topics associated with a specific line of research and their evolution [66]. It was based on the analysis of the manuscripts that compose the sample and, in

this study, was conducted with the support of the more suitable software for the analysis at an aggregate level: VOSViewer [67]. Bibliometric analysis is firmly established as a scientific specialty and is an integral part of research evaluation methodology, especially within the scientific and applied fields [68]. It consists mainly of bibliographic overviews of scientific contributions, often focused on a number of broad or more specialized subjects in the publishing field: geographical aspects (such as authors country affiliation, country under study, institution countries) [69], indicators of performance including publications' trend over time [70], subject domains or disciplines [71] or types of literature and authorships [72]. The analyses encompass data regarding different materials such as journal articles, books or conference proceedings. In order to extract and manipulate data, bibliometric methods are often performed through software [73], this is due to the helpful computerized methods but also to the fact that a bibliometric method has to include a certain volume of data in order to be statistically reliable [68]. The bibliometric method, performed through VOSViewer software, shows the result of a clustering process, a map which display a network composed by nodes and links between them. Particularly, in this study, we realized the co-occurrence map, in which each node is identified by a single term that recurs within the title and abstract of the selected dataset. Terms that have a high similarity (or rather compare together nearly within the title and abstract) should be located close to each other, while terms that have a low similarity should be located far from each other. The higher the similarity between two items, the higher the strongness of their links [74].

The overall strategy of the review methodology and related findings are illustrated in Figure 1.

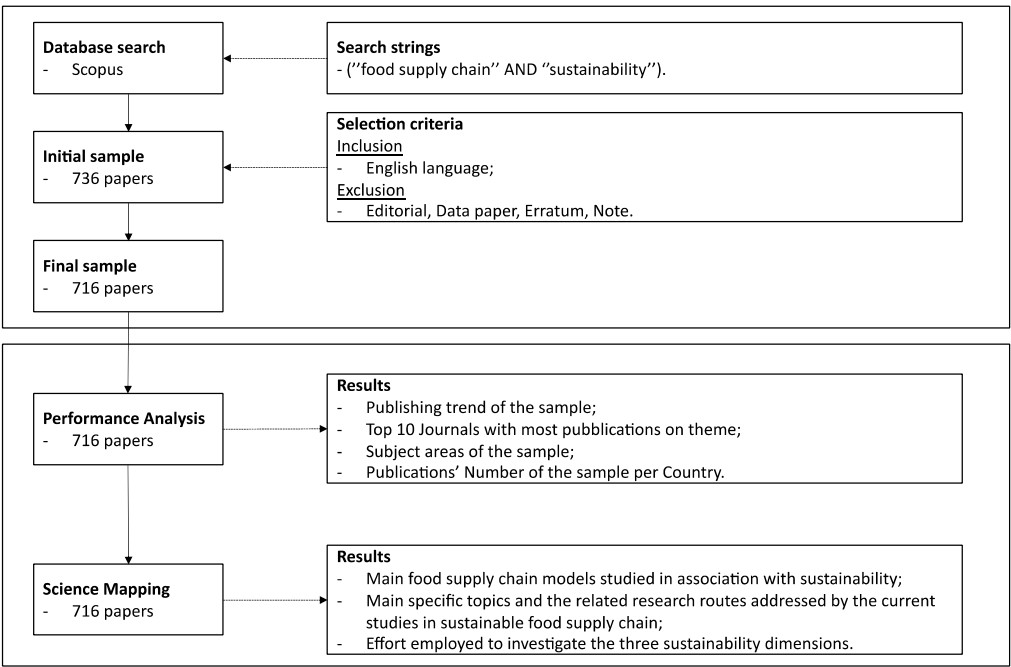

**Figure 1.** Review methodology.

In conclusion, several considerations about validity and reliability of the study are provided. Bibliometric methods introduce quantitative rigor into the subjective evaluation of the literature [66]. Validity explains how well the collected data cover the investigation area [75]. The quality of results from Science Mapping depends on several factors, such as the quality of keywords, the chosen database and the methods used for analysis [76]. To increase the validity of search terms, it is good practice to define appropriate keywords starting from the researchers' knowledge or involving a panel of scholars [77]. However, even when search keywords were carefully chosen, a database search could retrieve an

inappropriate sample since many journals' bibliographic data do not contain keyword or are affected by the indexer effect [66]. Therefore, a method to increase the validity consists of applying the search scheme in abstracts or full texts, in order to reduce the bias in keywords searching. Specifically, in this study, we adopted this strategy to increase the validity of the results at two different levels: (i) search scheme was applied in 'Article title, Abstract, Keywords' field of Scopus setting to retrieve the sample of analysis; (ii) Science Mapping was performed in the title and abstract of the analysis studies. Referring to the chosen data base, the recognized quality of Scopus helps to increase the validity of the results [62,63]. Finally, referring to the methods used for analysis, a term co-occurrence map, leveraging on the Scopus csv. Files, was performed extracting terms from the title and abstract field. The full counting method was chosen and a total of 16,494 recurrent terms could be considered for clustering process. To increase the validity of the results, a minimum number of occurrences of a term was set to 10 (term frequency level, tf = 10) and only 577 terms that meet this threshold were considered in the analysis. No one manual setting was performed to consider only the relevant terms.

## 3. Results: Performance Analysis

The Performance Analysis consists in describing the reviewed sample with some bibliometric information. This section starts by considering the publications trend over time, the top ten journals and the main subject areas discovered in the sample. Finally, the geographical distribution of the Institutions is described as well. These technical characteristics helped the researchers and practitioners in understanding the features of the research domain.

### 3.1. Publication Trend

The trends of publication about this topic start from the period 2003–2021, that was analysed. Indeed, the systematic literature review reports no evidence on papers published before the early 2000s. This likely happened because the agri-food digitalization process started in these years. As shown in Figure 2, the trend is low and almost constant in the first years (2003–2009). A moderate increase occurs between 2010 and 2013. Since 2014, the scientific landscape has been enriched, showing a higher but still constant growth trend, peaking in 2020 and 2021. Specifically, since the research was conducted in September 2021, we have reason to believe that the number of publications in 2021 will increase during the last part of the year. The positive trend reflects the overall interest in sustainability issues and, specifically, in the impact that these could have on food supply chain, which was covered in the Introduction section [4,6,17,19,22,23,26,30,33].

### 3.2. Top Ten Journals

A total of 127 different sources were identified in the analysis sample. Figure 3 shows the publication trend of the top 10 Journals by number of contributions: in this timeline graph, the *x*-axis lists the years of publication, and the *y*-axis represents the number of papers published by each journal in each year.

The sample analysed was referred to as the temporal line 2006–2021. However, only one paper was published in 2006 by the British Food Journal and then, from 2007 to 2011, agri-food publications suffered a setback. The years 2012–2013 saw a slight upturn in publications in this field of research. On the contrary, it is possible to highlight, globally, a growing interest in sustainability topics in the food supply chain and, consequently, an increase in publications from 2014 onwards with slight decreases only in 2015 and 2017.

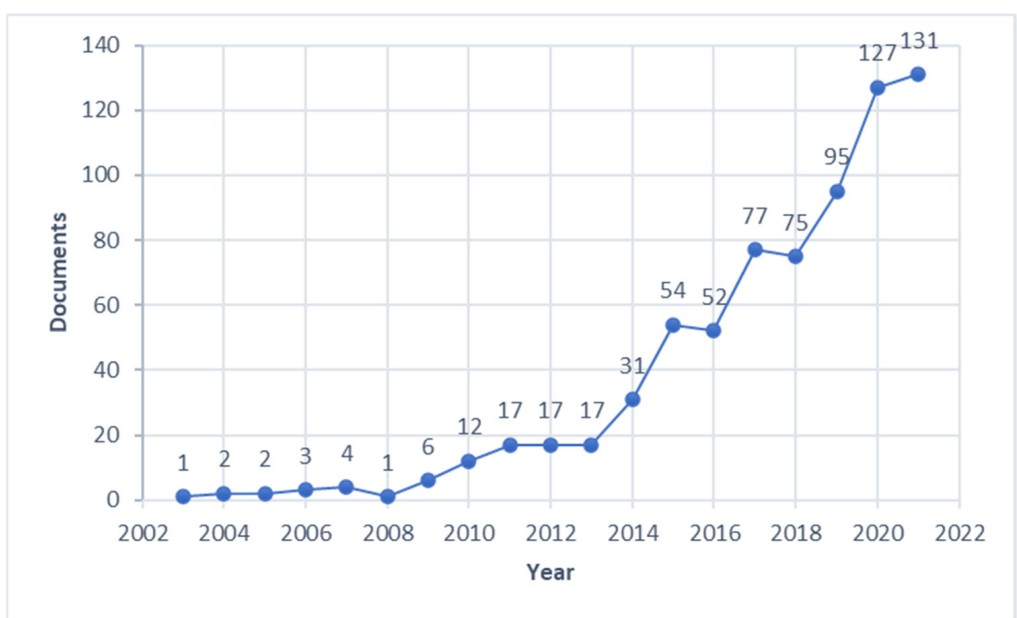

**Figure 2.** Publication trend of the sample.

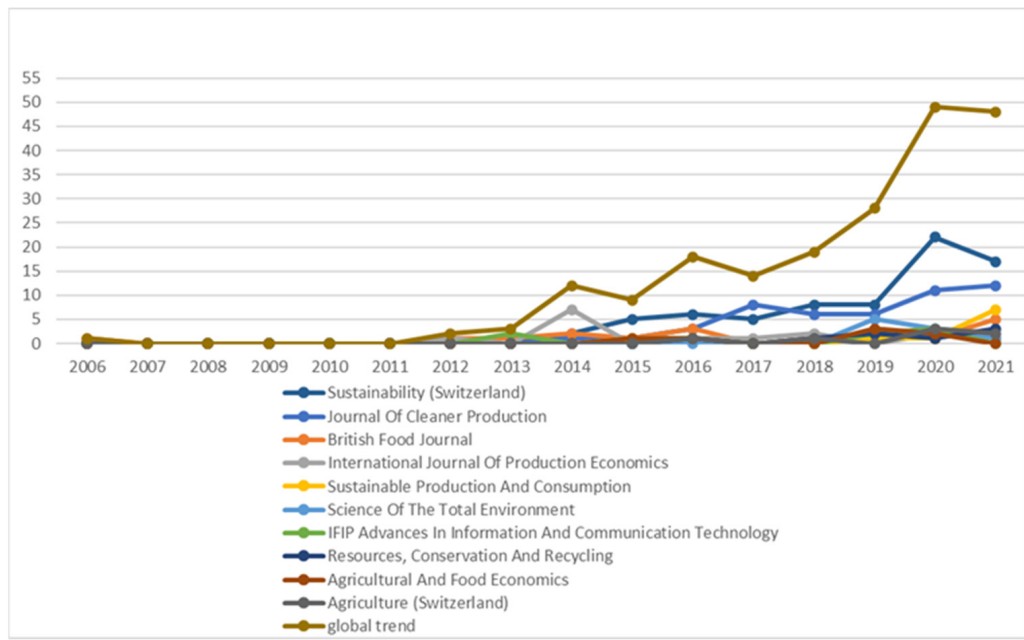

**Figure 3.** Publication trend of the top 10 Journals of the sample.

A global growth of publications occurs in the last two years (2020–2021), affecting, in particular, the Sustainability (Switzerland) Journal and the Journal of Cleaner Production.

Indeed, during the last three years (2019–2021), the Sustainability (Switzerland) Journal shows the highest publication rate, confirming the Journal's interest in sustainability and sustainable development also in the agri-food industry.

The upper brown curve represents the global trend of publications on agri-food sustainability during the period 2006–2021.

*3.3. Subject Areas*

The distribution of literature contributions for each subject area is shown in Figure 4. Specifically, in this analysis, we consider the subject areas declared by the journals in the sample. A total of 127 journals were found, which covers 22 different subject areas. "Agricultural and biological sciences" is the most recurrent subject area (47 occurrences),

followed by "Environmental science" (39 occurrences) and "Business, management and accounting" (29 occurrences). These main subject areas presented in this histogram covered, respectively, 17%, 14% and 11% of the total amount of contributions.

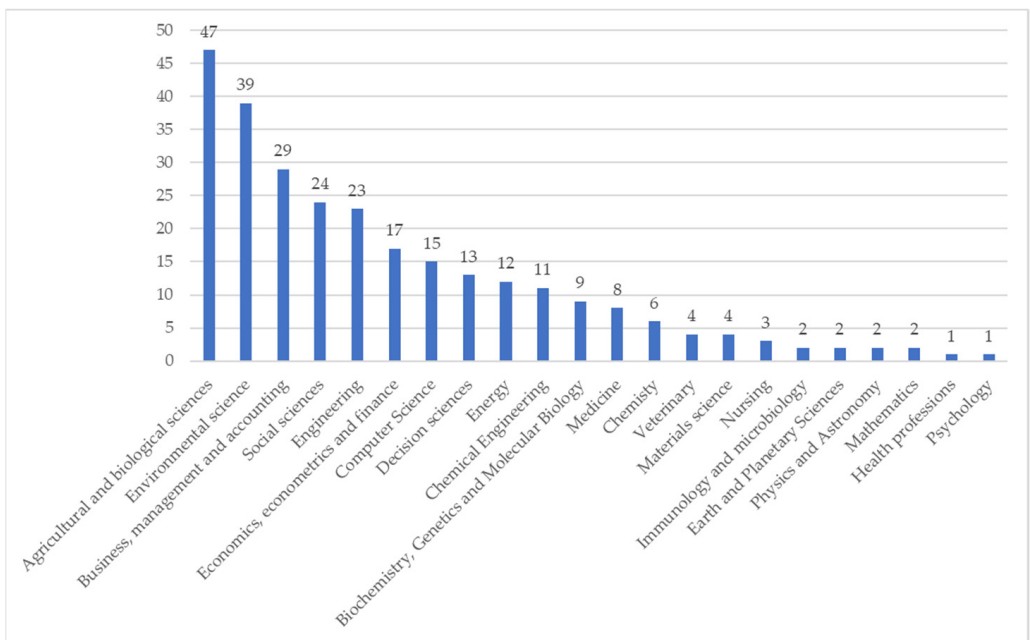

**Figure 4.** Subject areas distribution of the sample.

The subject areas of "Social sciences" and "Engineering" follow immediately afterwards, with 24 and 23 occurrences, respectively. The subject areas that showed the lowest number of occurrences belong to "Veterinary", "Mathematics", "Physics", "Health professions" and "Psychology" fields.

### 3.4. Geographic Distribution

The topic of sustainability in the agri-food supply chain has resulted in global interest. Figure 5 shows the geographical distribution of the studies referred to by the countries of the authors' affiliation body. It is interesting to note that European and American territories are more interested in the topic. Especially, the United Kingdom (UK) and Italy represent the most productive Countries counting, respectively, 156 occurrences and 146 occurrences, followed by the United States (101 occurrences). Since much of the literature reviewed comes from Europe and the United States, it is clear that these continents have had resources to undertake research about the food supply chain. Indeed, in these territories are located the majority of food giants, which were the pioneers in starting food supply chain projects embraced as valuable case studies from the European and American universities and research institutions.

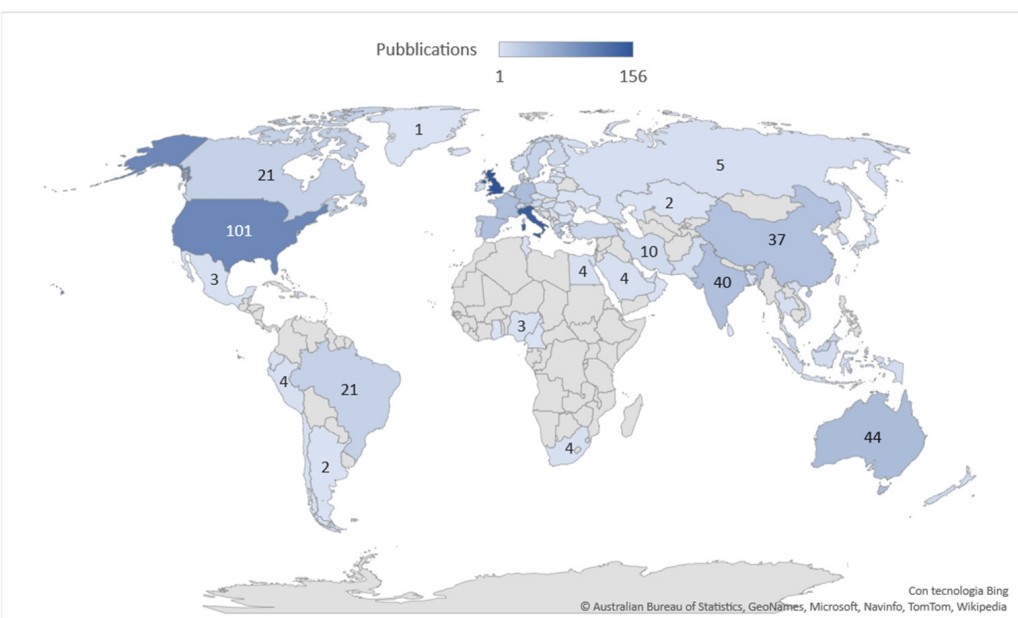

**Figure 5.** Geographic distribution of the sample (Institutions' countries).

Moreover, the literature review mirrors cultural, social and governmental aspects about food systems and sustainability that concern Italy, the UK and the United States. Italy is recognized around the globe thanks to the "Made in Italy" brand that sees food as a cultural and traditions protagonist [78]. Specifically, the Italian food industry is the first export sector of "Made in Italy" in the UK, in terms of volumes, although it is now paying the highest bill due to Brexit [79]. On the contrary, "Made in Italy" food export intercepts an increment of +15.9% by Euro value for export in the United States, putting Italy above the other international competitors [80]. Therefore, the food supply chains have more impact on the economy, the culture and the environment of these countries, gaining attention from consumers and government. On May 2020, the European commission presented the "farm to fork" strategy with the aim of building a sustainable food system, to safeguard food security and protect European citizens and nature [31]. In the same line and in the same period, the United States Environmental Protection Agency (EPA), United States Department of Agriculture (USDA), and Food & Drug Administration (FDA) built new strategies toward sustainable food system progress [32].

## 4. Results: Science Mapping Analysis

Science Mapping was performed through cluster analysis. The clustering process, concluded by VOSViewer software, supplies the thematic map of the research domains (food supply chain and sustainability). The present analysis provides a series of graphical maps (from Figures 6–13) that represent the overall network of recurrent terms, and, in sequence, the zoom on the main node of each cluster. Specially, Figure 6 shows the resulting overall thematic map, characterized by a network of seven clusters, in which, a node corresponds to a recurring term. Each cluster is marked by a different color and ordered by numerosity of nodes: the first cluster (149 nodes) in red, the second (105 nodes) in green, the third (104 nodes) in blue, the fourth (84 nodes) in yellow, the fifth (52 nodes) in lilac, the sixth (48 nodes) in light blue and the seventh (35 nodes) in orange. This map arranges terms by occurrences and allows exploration, cluster by cluster, of the terms and interpretation of the related common themes. Specifically, the interpretation of common themes is conducted on the basis of the researchers' knowledge and experience, that, fostered by the software capabilities, permits carefully obtaining results and finding answers to the research questions, as defined in the Methodology section [67]. Thus, in order to individuate (i) the main research routes in sustainable food supply chain, (ii) the main supply chain models studied in association with sustainability and (iii) the effort

made by the academia in analysing the several sustainability dimensions, each cluster was analysed. Other considerations could be made leveraging the network density map and the overlay network map.

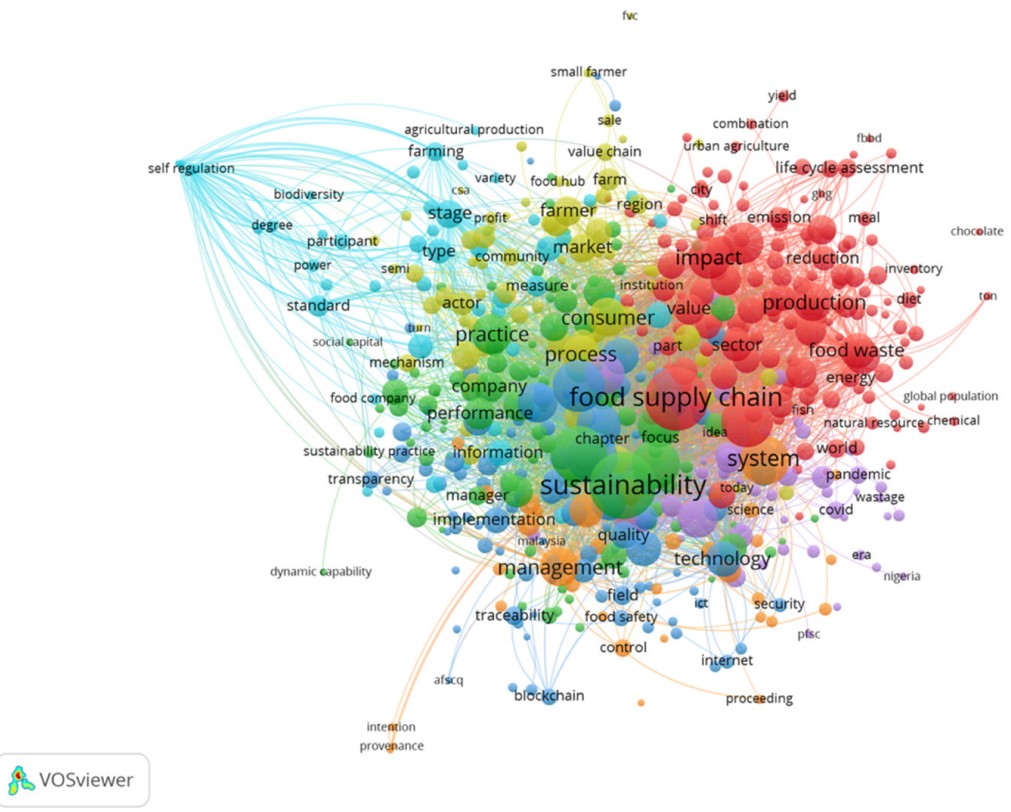

**Figure 6.** Thematic map resulting from VOSViewer clustering process.

### 4.1. Cluster Analysis

Cluster 1, the red one, is the biggest of the networks and encompasses 149 terms, captained by the "food supply chain" node, which is one of the research keywords. This node recurs 681 times in the data set and has 574 links within the network.

The "food supply chain" node, in combination with other similar nodes such as "entire food supply chain" and "global food supply chain", guides the common theme of Cluster 1 that seems to collect terms linked to the food supply chain domain. Specifically, different elements of the supply chain and related concepts are mentioned: "delivery", "cost", "distribution", "efficiency", "effort", "feasibility", "food manufacturing", "food production", "food supply", "food system", "global supply chain", "household", "impact", "impact category", "increase", "lifecycle", "life cycle assessment", "order", "output", "packaging", "parameter", "phase", "place", "plant", "potential", "processing", "raw material", "resource", "sector", "shift", "source", "storage", "supply", "sustainable food system", "transport", "transportation", "treatment" and "value". Moreover, examining terms, it is possible to note that sustainability is addressed considering the environmental dimension, with particular attention to the evaluation of the impact that all phases of the entire food life cycle have on the environment, as attested by: "carbon emission", "carbon footprint ", "emission", "energy", "energy consumption", "environmental impact", "environmental performance", "environmental sustainability", "food loss", "food waste", "greenhouse gas emission", "ghg emission", "impact", "impact category", "life cycle assessment", "lca", "natural resource", "parameter", "plate waste", "reduction", "sustainable food system", "sustainable consumption", "waste", "waste reduction", "water" and "water footprint". It is interesting to note that Cluster 1, unlike the others, encompasses more terms determining specific food supply chains in terms of typology of food supplied: "animal", "chocolate", "fish", "fruit", "meat", "vegetable" and "fresh produce". This evidence leads researchers to

assume that the evaluation of the environmental impact of food lifecycle within the supply chain is more investigated considering particular food product categories.

With reference to the "food supply chain" node, in addition to the internal links with other nodes of the same cluster, there are links to others clusters, which gives an input to interpret the association among different specific topics. Specifically, Figure 7 represents the main view of the "food supply chain" node and shows its linkages with terms of the cluster to which it belongs (e.g., "production", "sector", "impact", "food waste", "cost", "consumption") and also with terms belonging to other clusters (e.g., "sustainability" and "practice" in the green cluster; "study" and "technology" in the blue cluster; "system" and "management" in the orange cluster; "consumer" and "market" in the yellow cluster; "standard" and "self regulation" in the light cluster; "challenge" and "strategy" in the lilac cluster).

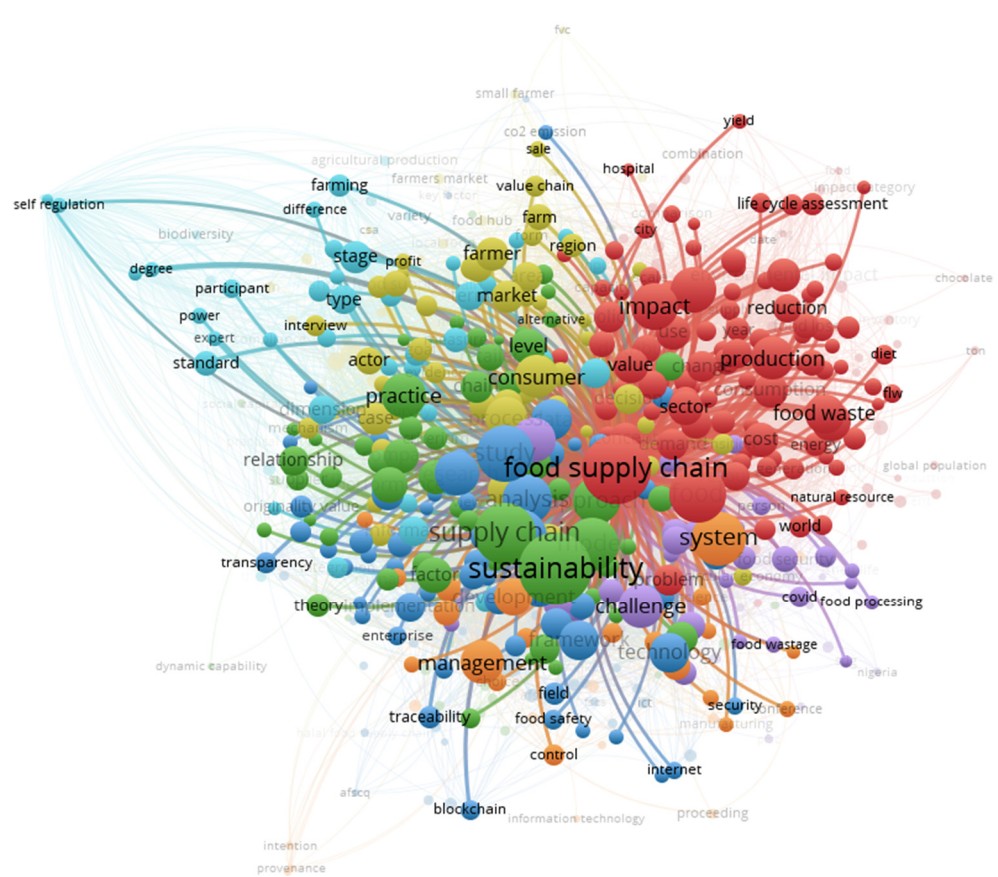

**Figure 7.** "Food supply chain" node and its linkages.

Cluster 2, the green one, is composed of 105 terms guided by the "sustainability" node, which is one of the search keywords. This node recurs 738 times in the data set and results in 571 links within the network. Figure 8 represents the main view of the "sustainability" node and shows its linkages with terms of its own cluster (e.g., "practice", "supply chain", "relationship") and also with the terms belonging to other clusters (e.g., "food supply chain", "food waste" in the red cluster; "study", "technology" in the blue cluster; "system", "management", in the orange cluster; "consumer", "market" in the yellow cluster; "standard", "farming" in the light cluster; "challenge", "strategy" in the lilac cluster). This main node gives the common print of Cluster 2, which seems to collect terms linked to the sustainability domain. Particularly, the three sustainability dimensions are directly or indirectly mentioned: (i) the environmental one is represented, among others, by "environmental sustainability", "environmental impact", "environmental performance", "carbon emission", "carbon foot print" terms; (ii) the social one is represented, among

others, by "social sustainability", "social impact", "corporate social responsibility" terms; finally, (iii) the economic one is represented, among others, by "economic sustainability", "performance", "success", "sustainable business model" terms. No particular inclination to food domain is attested, in fact the second main node is "supply chain", a general term linked to "industry" in general. The common theme of Cluster 2 arose from the analysis of recurring terms (e.g., "applicability", "approach", "assessment", "attention", "best practice", "business", "case study", "company", "constraint", "criterium", "decision maker", "decision making", "driver", "key driver", "governance", "improvement", "implication", "investment", "manager", "policy maker", "practice", "progress", "roadmap", "sustainability assessment", "sustainability indicator", "sustainability performance", "sustainability practices", "sustainable business model", "sustainable supply chain", "sustainable supply chain management") and is related to environmental, social and economic sustainability dimensions along the supply chain, and is, as well, regarded as key drivers for adoption of best practices in real business context.

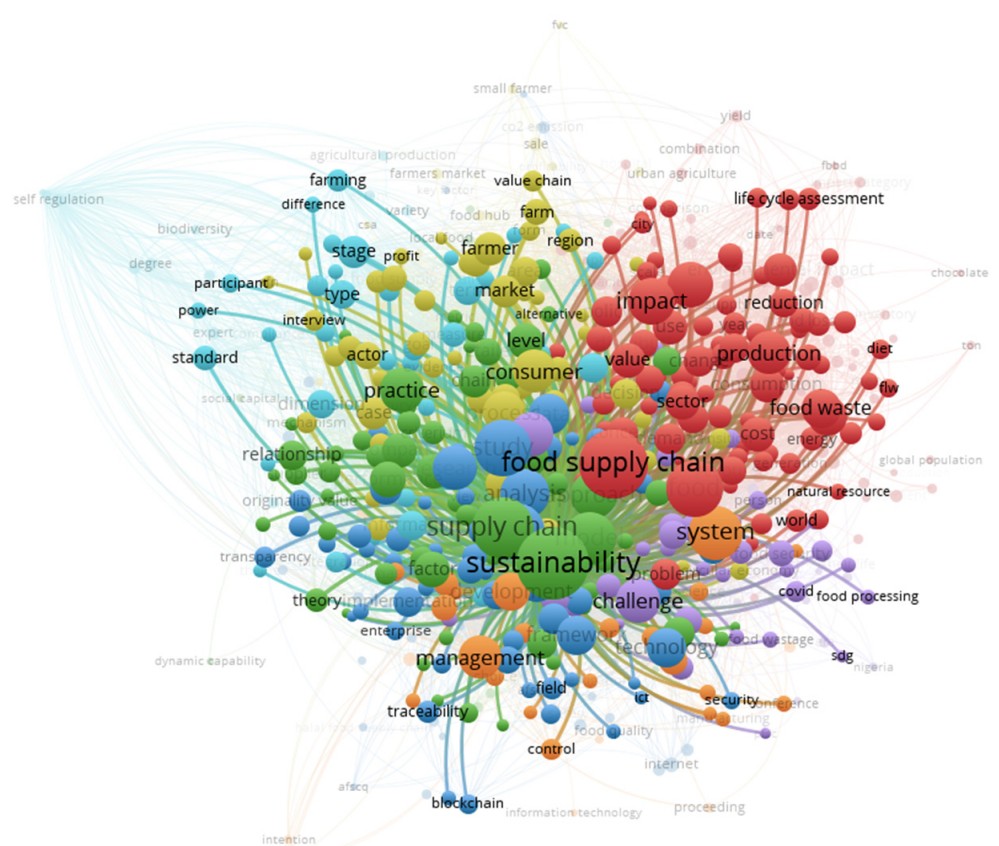

**Figure 8.** "Sustainability" node and its linkages.

Cluster 3, the blue one, is composed of 104 terms. The main node is "study" (recurring 457 times and with 745 linkages) followed by "analysis" (recurring 342 time and with 566 linkages) and "research" (recurring 270 time and with 561 linkages) (Figure 9). These terms give, to Cluster 3, the print of empirical research. Specifically, the presence of nodes such as "technology", "enabler", "analysis", "blockchain", "blockchain technology", "communication technology", "conceptual framework", "data collection", "empirical research", "ict", "implementation", "information sharing", "internet","iot", "origin", "security", "solution", "traceability", "traceability system", "transparency", "trust" and "quality" connotes the empirical research theme as oriented to technological solutions for agri-food system able to face traceability, transparency, security, trust and quality issues. Therefore, the application domain of Cluster 3 is the agri-food system, as attested by the following nodes:

"agrifood supply chain", "afsc", "agrifood sector", "food quality", "food safety", "food supply chain management", "fresh food supply chain", "fresh produce", "fsc", "halal food supply chain", "sustainable agrifood supply chain" and "sustainable agriculture". In summary, Cluster 3 seems to be oriented to empirical research or studies about technological solutions able to assure food quality and safety, traceability and to enhance trust.

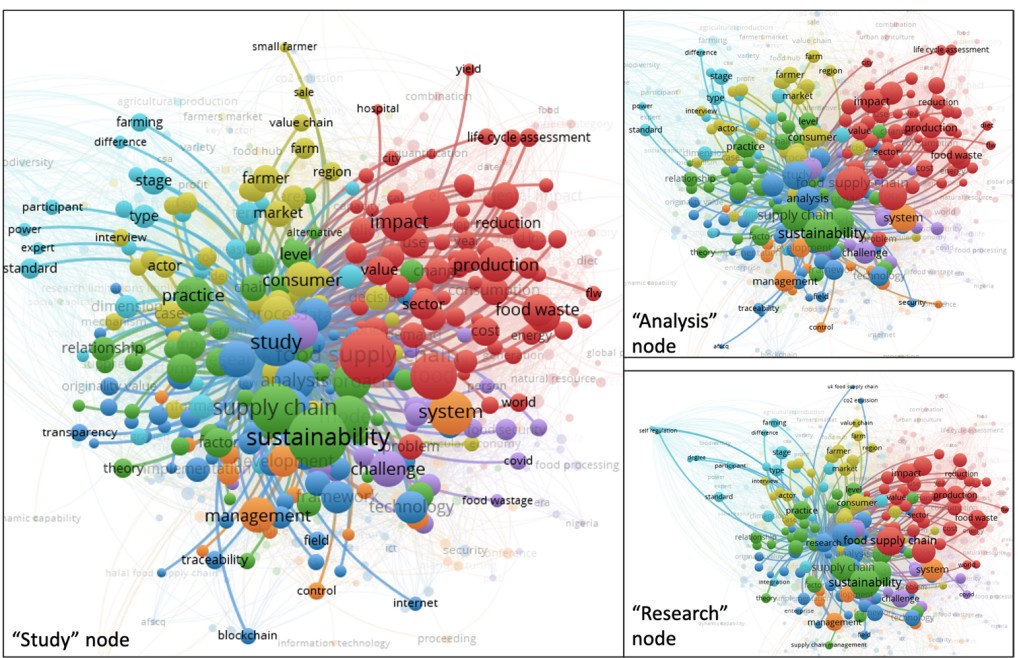

**Figure 9.** "Study", "analysis" and "research" nodes and their linkages.

Cluster 4, the yellow one, is composed of 84 items and shows the "consumer" (recurring 261 times and with 559 linkages) and "process" (recurring 236 times and with 558 linkages) nodes as main ones. These main nodes and the related links are represented in Figure 10. Investigating through the other nodes, several well-focused terms capture the research attention: "actor", "advantage", "agri food system", "agri food supply chain", "alternative food network", "circular economy", "community", "consumer", "cooperation", "customer", environmental benefit", farm", "farmer", "farmers market", "food chain", "food hub", "food producers", "food sustainability"; "Italy", "local food", "local food supply chain", "local food system", "value chain", "market", " profitability", "sfsc", "sfscs", "short food supply chain", "smaller farmer", social benefit", "social sustainability", "spain", "sustainable development" and "turkey". The sustainability issue is considered from all of its three viewpoints. The agri-food sector, with particular focus on the short and local food supply chain models, results in the main domain. It is interesting to note the presence of nodes with country names, helpful to give a geographical context of the Cluster 4 common theme. In summary, this cluster encompasses terms linked to the short and local supply chain models capable of involving the consumers in the supply chain and enabling the environmental and social sustainability of small farmers. Specifically, the opportunity to encompass the consumer in the food lifecycle enables the transformation of the supply chain in a value chain.

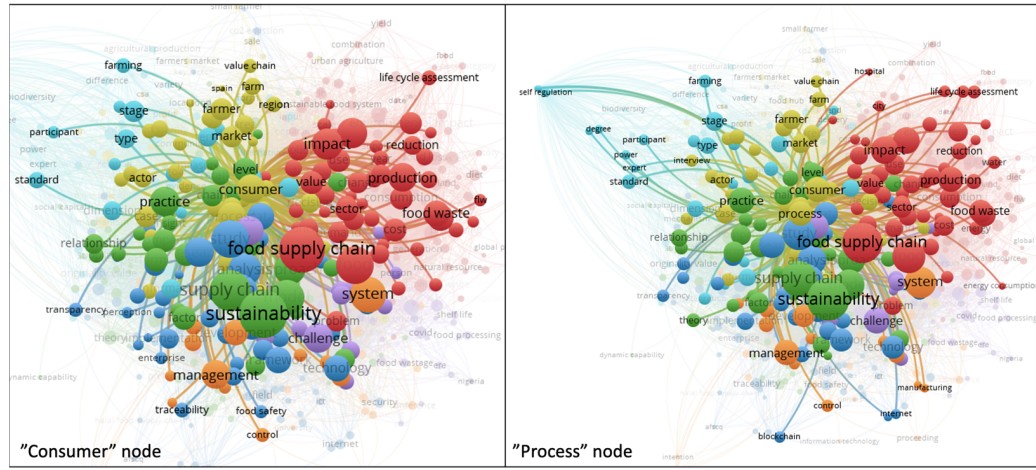

**Figure 10.** "Consumer" and "process" nodes and its linkages.

Cluster 5, the lilac one, is composed of 52 items and the main nodes are "strategy" (recurring 270 times and with 548 linkages) and "challenge" (recurring 265 times and with 542 linkages) which characterize the common theme of Cluster 5 (Figure 11). This cluster demonstrates a focus on the challenges that the world is facing today and the related strategies. More clarity derives from the analysis of the others nodes, leading the researcher to individuate a specific focus of the common theme: "availability", "challenge", "competitiveness", "covid", "crisis", "digital technology", "disruption", "economic sustainability", "economy", "food insecurity", "health", "infrastructure", "logistic", "need", "organization", "pandemic", "perishable food supply chain", "perishable product", "risk", "safety", "shelf life", "strategy" and "sustainable development". Technology is also mentioned in this cluster, although it is not the principal focus, as happened in Cluster 3. Therefore, the common theme of Cluster 5 is the attention to the strategies, with a focus on technological solutions for facing current challenges and crises (economic, environmental and organizational in the perishable food supply chain) such as the pandemic condition caused by covid 19 global diffusion.

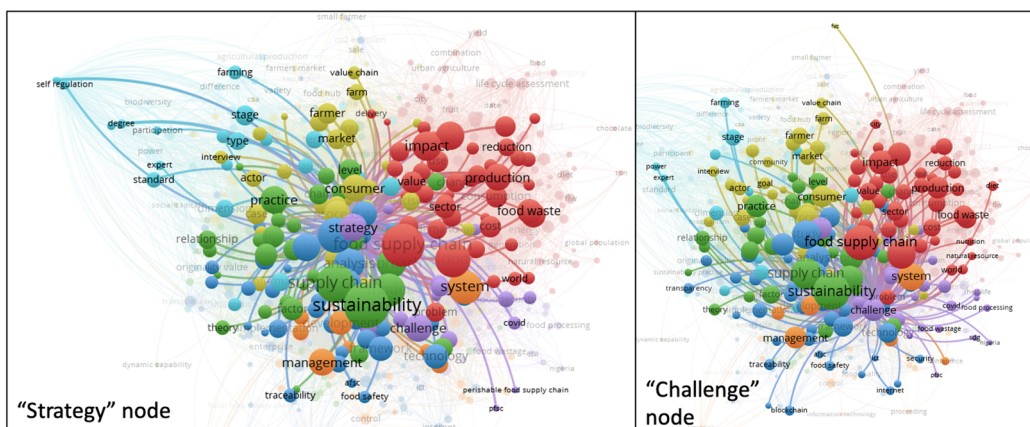

**Figure 11.** "Strategy" and "challenge" nodes and their linkages.

Cluster 6, the light blue one, is composed of 48 items with "stage" (recurring 133 times and with 505 linkages) as a main node. This main node, although it has a low level of occurrence in comparison with the main nodes of other clusters, is widely connected with other clusters, as Figure 12 shows. Analyzing the nodes of Cluster 6 (e.g., "account", "agrobiodiversity", "biodiversity", "compliance", "degree", "difference", "effectiveness", "evaluation", "expert", "farming", "food company", "government", "industry self regulation", "integration", "measure", "monitoring", "regulation", "reliability", "self regulation"

and "standard"), researchers individuate a clear connotation toward standards or regulations in the agri-food sector in association with environmental sustainability issues linked to agrobiodiversity. Therefore, the common theme of Cluster 6 results as the analysis of food supply chain compliance to regulations and standards with particular focus on farming stage and with the aim to safeguard agrobiodiversity.

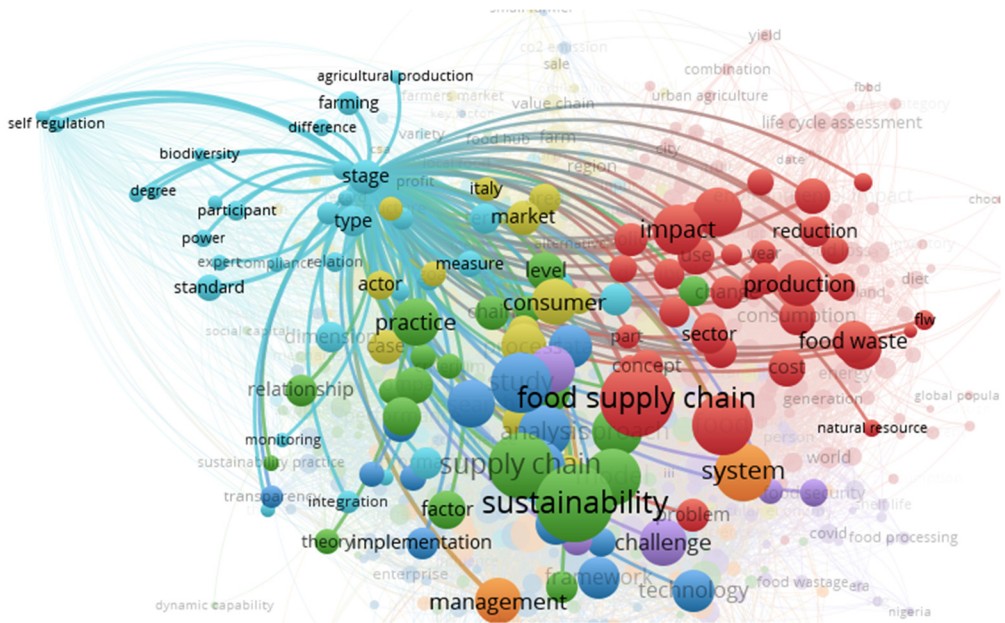

**Figure 12.** "Stage" node and its linkages.

Cluster 7, the orange one, is composed of 35 items; although it is the most minor cluster in terms of nodes numerosity, it shows several big nodes, among which are "system" (recurring 450 times and with 557 linkages), "management" (recurring 256 times and with 546 linkages) and "development" (recurring 203 times and with 536 linkages). This last cluster seems to be oriented to the knowledge produced along the activities of investigation, education, management, application, planning, dissemination and development. This evidence emerges from terms analysis, such as: "application", "attitude", "building", "conference", "development", "education", "idea", "investigation", "knowledge", "management", "manufacturing", "optimization", "planning", "questionnaire", "science", "simulation" and "university". The domain is, in this case, not clearly interpretable.

As previously stated, Cluster 7 encompasses few nodes but has a large size. This evidence means that some terms are more recurrent within the data set and intercept other clusters through strong linkages. Figure 13 shows how "system", "management" and "development" nodes are linked with other ones. Specially, "system" is connected to: (i) Cluster 1 with "food", "food supply chain", "product", "production", "value", "environmental impact", "food waste" and "policy"; (ii) Cluster 2 with "sustainability", "supply chain", "approach", "model", "practice" and "factor"; (iii) Cluster 3 with "paper", "analysis", "data", "framework" and "technology"; (iv) Cluster 4 with "consumer" and "process"; Cluster 5 with "challenge", "operation" and "strategy"; (v) Cluster 6 with "information". The "management" node is connected to: (i) Cluster 1 with "food supply chain" and "product"; (ii) Cluster 2 with "sustainability", "approach", "model", "practice" and "factor"; (iii) Cluster 3 with "study", "research", "literature", "analysis", "framework" and "data"; (iv) Cluster 4 with "process"; (v) Cluster 5 with "challenge"; (vi) Cluster 6 with "stage" and "self regulation". The "development" node is connected to: (i) Cluster 1 with "food supply chain" and "food"; (ii) Cluster 2 with "sustainability", "supply chain", "approach" and "model"; (iii) Cluster 3 with "research", "study", "analysis" and "paper".

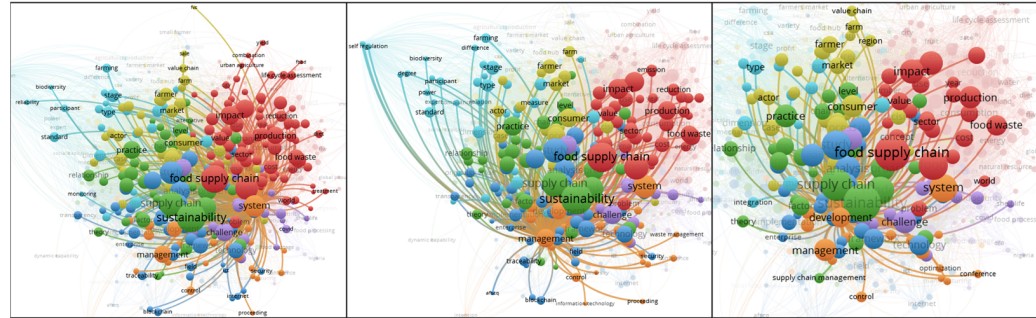

**Figure 13.** "System", "management" and "development" nodes and their linkages.

*4.2. Density Visualization*

The density visualization of the thematic map, in Figure 14, is a particular view of the network leveraging on density parameter. Specifically, the denser a specific zone of the network, the brighter and more yellow the zone. Figure 14 shows a major density of the network around the two search keywords "food supply chain" and "sustainability". This result is not surprising because, due to the setting of the search scheme, these keywords compare in the title, abstract and/or keywords of the data set. The terms that are strictly satelliting around these two nodes are more linked with them, meaning that they are frequently found together within a manuscript. These terms are "process", "consumer", "system", "impact", "management", "technology", "company", "performance", "practice", "production", "information" and "quality".

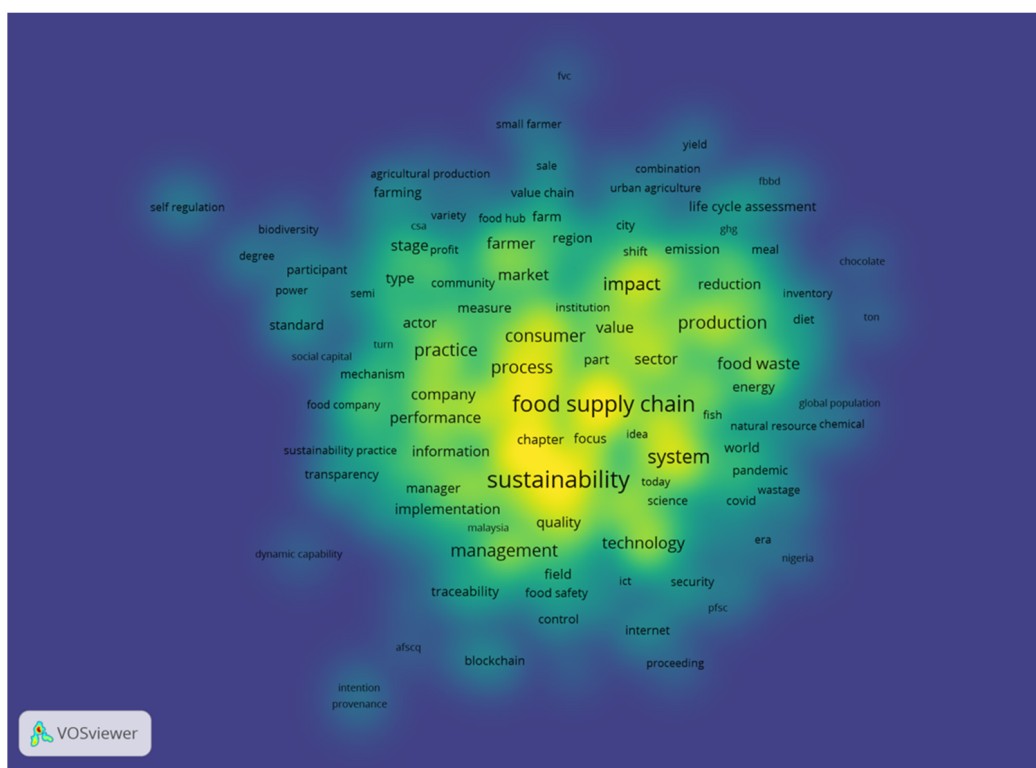

**Figure 14.** Density visualization of the thematic map.

*4.3. Overlay Visualization*

The overlay visualization shows the evolution of terms comparing over time (Figure 15). This result gives a part of knowledge about the actuality of themes that are represented in this map in yellow color. For example, blockchain results as one of the more recent concepts within the network, and this result is confirmed by the fact that the blockchain

is one of technology deriving from the Industry 4.0 paradigm. It is front line in more industrial sectors, including the agri-food one. Another piece of evidence is the "covid" and "pandemic" location: obviously nowadays. However, the main nodes of the network, which corresponding to the chosen search keywords, appeared within the dataset since the 2017. Otherwise, Cluster 6 encompasses terms related to regulation and standard topic, resulting in the oldest ones.

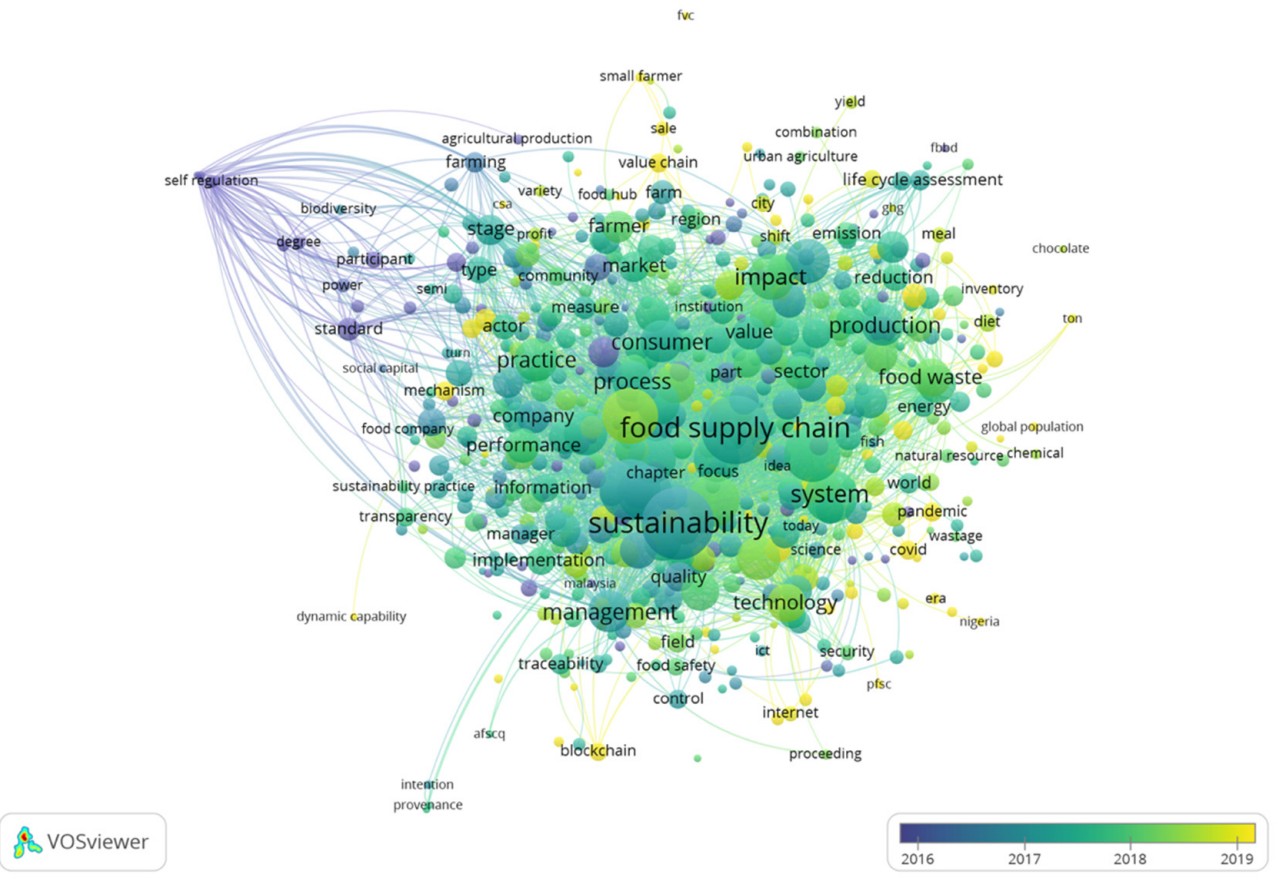

**Figure 15.** Overlay visualization of the thematic map.

## 5. Discussions

In this section, we debated how this literature review is useful in providing answers to the research questions investigated. Therefore, considering the insights that emerged from Descriptive and Thematic analysis, the answer to the research question—"What are the main research routes addressed by the studies on sustainable food supply chain?"— emerges. From the analysis of the recurring subject areas, it emerges that "agricultural and biological sciences", "environmental science" and "business, management and accounting" are the most recurrent subject areas involved in the studies of sustainable food supply chains. Thematic analysis, through the definition of thematic clusters, allows us to explore more deeply the topics that build this area of research, discovering the presence of six research routes that academia addressed in the sustainable food supply chain studies: (i) investigation of food supply chain, (ii) investigation of agri-food sustainability, (iii) investigation of technological solutions for sustainable agri-food, (iv) investigation of consumer and small farmer protection, (v) investigation of agri-food challenges and strategies, and (vi) investigation of agri-food standards and regulations. Focusing on the overlay visualization of the thematic map, it is interesting to note that the investigation of agri-food standards and regulations is the most mature with most of the terms placed in 2016. On the contrary, the investigation of agri-food challenges and strategies is the

youngest research route, with most of the terms placed between 2019 and 2021. Interesting to note is the time position of the investigation of technological solutions for sustainable agri-food research routes, which appears mature for the same terms (e.g., traceability system, communication technology) and young for others (e.g., blockchain, IoT, digital technology), confirming the evolution of technologies over time. The timing position of the research routes is summarized in Figure 16.

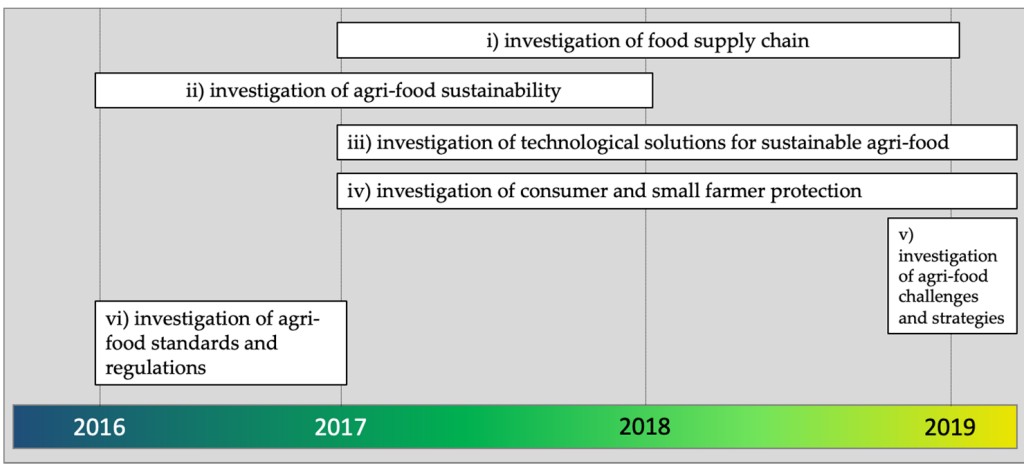

**Figure 16.** Timing evolution of the emerging current research routes.

With the aim to provide more insights about the theme explored in these discovered trends, Table 1 proposes, for each current research routes, a list of specific themes that emerged from the analysis.

**Table 1.** Current research routes and related research themes.

| Current Research Routes | Main Research Themes |
|---|---|
| (i) Investigation of food supply chain | Searching efficiency in food supply chain. |
| | Global food supply chain management. |
| | Food lifecycle assessment. |
| | Environmental impact of the food supply chain. |
| | Raw materials and resource management along the food supply chain. |
| (ii) Investigation of agri-food sustainability | Environmental impact of agri-food industry. |
| | Social impact of agri-food industry. |
| | Corporate Social Responsibility in agri-food industry. |
| | Economic impact of agri-food industry. |
| | Sustainable business model for agri-food industry. |
| (iii) Investigation of technological solutions for sustainable agri-food | Technological enabler to sustainable agri-food industry. |
| | Food traceability systems. |
| | Blockchain technologies in agri-food industry. |
| | Information sharing and communication technologies in agri-food industry. |
| | IoT technologies in agri-food industry. |

**Table 1.** *Cont.*

| Current Research Routes | Main Research Themes |
|---|---|
| (iv) Investigation of consumer and small farmer protection | Alternative food network analysis. |
| | Circular economy models. |
| | Cooperation analysis among farmers, producer and consumers. |
| | Local food system models. |
| | Profitability and social benefits for agri-food communities. |
| (v) Investigation of agri-food challenges and strategies | Agri-food competitiveness. |
| | Impact of COVID19 pandemic in agri-food industry. |
| | Food insecurity and risk management. |
| | Shelf life and perishable product management. |
| | Infrastructure and digital technology management. |
| (vi) Investigation of agri-food standards and regulation | Industry self-regulation. |
| | Food standard management. |
| | Food supply chain compliance to regulations and standard. |
| | Impact of government and measure on agri-food industry. |
| | Safeguard of agrobiodiversity. |

Leveraging on the findings provided by the density visualization map of the terms, we can conclude that the research field of sustainability in food supply chains is focused on management issues capable of generating impacts on processes, systems, practices, production and quality. The sustainable development in the food supply chain leverages on technologies and information management with the aim to improve the company performance, the product quality and the consumer satisfaction. Considering the importance of the food industry and its role in society, economy and environment, the emerging current research routes are worthy of further development in the future. For example, the investigation of food standards and regulations could find new life in the analysis of the role that international agri-food certifications play in sustainable development.

Referring to the research question—"What are the main food supply chain models studied in association with sustainability?"—it is interesting to note that, leveraging on the cluster analysis results, we found several emergent food supply chain models studied in association with sustainability: "fresh food supply chain" and "halal food supply chain" from Cluster 3; "local food supply chain" and "short food supply chain" from Cluster 4; "perishable food supply chain" from Cluster 5. Moreover, from Cluster 1, the presence of specific food products ("animal", "chocolate", "fish", "fruit", "meat", "vegetable") emerges, even if they do not appear in association with the concept of supply chain (e.g., we found "fish" and not "fish supply chain"). The relations that these models have with sustainability issues could be retrieved from the network analysis of each cluster. Fresh food supply chain and halal food supply chain models appear to be studied from a technological viewpoint with the aim to enable the agri-food supply chain to a wide concept of sustainability that encompasses transparency, traceability, safety, quality and trust issues. Local food supply chain and short food supply chain are models capable of fostering economic and environmental sustainability, promoting $CO_2$ and waste reduction through the circular economy approach and reinforcing social sustainability through the protection of small and local food businesses. It is interesting find these models related to specific countries, such as Spain, Italy and Turkey. Probably, this is related to the importance that local food and local producers have in the culture and traditions of these countries. The importance

of the theme for Italy is also confirmed by the geographical distribution of the sample, in which this country has the second position in manuscripts production.

The perishable food supply chain model studied, contextually, safety and risk issues probably amplified by the pandemic ("covid", "food insecurity", "health", "risk", "safety"). The research around this model is focused on a sustainable development path able to consider economic and social dimensions of sustainability.

Finally, looking for the breadth of research along the three dimensions of sustainability (Research question—"Is one of the sustainability dimensions more investigated than the other ones?"), several considerations can be made. The studies focused on sustainability issues, represented by the themes in Cluster 2, addressed sustainability considering the three dimensions: (i) environmental sustainability, which is represented by "environmental impact", "environmental performance", "environmental sustainability", "carbon emission" and "carbon foot print" concepts; (ii) social sustainability, which is represented by "social sustainability", "social impact" and "corporate social responsibility" concepts, (iii) the economic sustainability, which is represented by "economic sustainability", "performance", "success" and "sustainable business model" concepts. The same holistic approach is retrieved in Cluster 4 which, focusing on consumer and small producer protection, faces the sustainability issue from several viewpoints: environmental in the promotion of local supply chain able to decrease the environmental impacts related to the long distribution chain (e.g., carbon footprint), social in the promotion of local food ensuring better performance for local and small farmers and economic in the promotion of strategies of circular economy and waste reduction and in the transformation of supply chain in value chain.

However, from the analysis of Cluster 1, it emerges that the food supply chain issues are focused on the environmental sustainability dimension, paying attention to emission, footprint, waste, natural resource and applying lifecycle assessment methodology to evaluate the impact that the production, distribution and selling activities of several kinds of products ("animal", "chocolate", "fish", "fruit", "meat", "vegetable") generate on the environment. Similarly, Cluster 5 focuses on several agri-food challenges and strategies, addressing elements of social and economic sustainability. Social sustainability in this cluster is represented by several terms related to food safety, insecurity and risk. Economic sustainability appears represented, along with other terms, by "crisis", "competitiveness" and "economy". Probably, the presence of the COVID-19 pandemic has moved the research focus to social and economic challenges that, in health and wellness conditions, would have been less urgent to solve.

With the aim of providing a comprehensive analysis of the research breadth of the three dimensions of sustainability, considerations about their level of maturity were made. Figure 17 shows some of the most representative terms of the sustainability dimensions retrieved in the terms network, placing them along a temporal frame. Specifically, economic sustainability terms were shown in blue, environmental sustainability terms in green and social sustainability terms in orange. Moreover, the researchers reported in white some terms, which, according to their opinion, are capable of encompassing contemporarily the three sustainability dimensions.

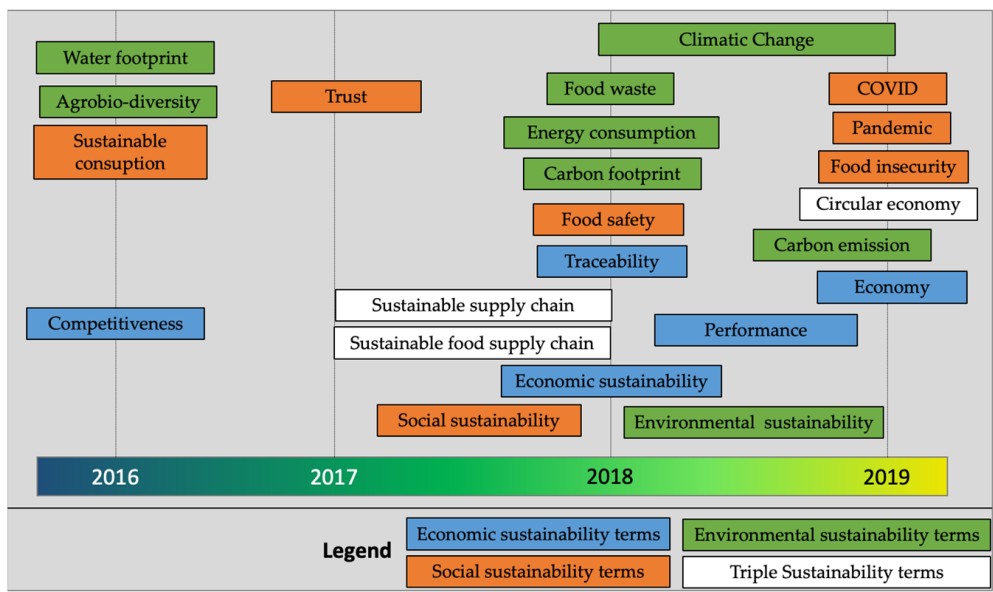

**Figure 17.** Time visualization of sustainability related terms.

Although the terms "economic sustainability", "social sustainability" and "environmental sustainability" appeared in the network between 2017 and 2018, other representative terms of these sustainability dimensions were found in the entire period of time considered (before 2016–after 2019). This evidence led the researchers to assert that the three dimensions were equally addressed during the time.

## 6. Conclusions, Implications and Limitations

Over the last few decades, researchers have paid considerable attention to sustainability practices along the food supply chain, especially in relation to the current challenges in the food industry such as: world hunger, climatic change, environmental pollution, food frauds, food safety, loss of biodiversity, pandemic, rural protection, trust and waste management. Several studies aimed to analyze sustainability issues in the food supply chain. With the aim of systematizing the knowledge in this research field and investigating three research questions, this study proposed a systematic literature review with bibliometric analysis. A sample of 716 papers was analyzed firstly from performance viewpoint, identifying the publications trend, the list of top ten journals, the main subject areas and the geographical distribution of the sample. After that, performing a Science Mapping analysis, seven clusters of terms were detected. The analysis of these clusters allowed for the identification of six current research routes: (i) investigation of food supply chain, (ii) investigation of agri-food sustainability, (iii) investigation of technological solutions for sustainable agri-food, (iv) investigation of consumer and small farmer protection, (v) investigation of agri-food challenges and strategies and (vi) investigation of agri-food standards and regulations. Moreover, five emergent food supply chain models studied in association with sustainability were discovered in the current literature: (i) fresh food supply chain, (ii) halal food supply chain, (iii) local food supply chain, (iv) short food supply chain and (v) perishable food supply chain. Finally, the breadth of research along the three dimensions of sustainability (economic, environmental and social) was discussed. We can conclude that the research field of sustainability in the food supply chain is focused on management issues capable of generating impacts on process, systems, practices, production and quality.

### 6.1. Implications

Leveraging on the discussion of the findings of this study, several theoretical and practical implications emerged. The results of the study could benefit researchers, academics and practitioners, helping them to understand the multiplicity of concepts that compose

the research field of the sustainable food supply chain. To the best of our knowledge, this study is the first to provide a systematic overview and critical appraisal of the extant literature on sustainable food supply chain. Specifically, the six current research routes, the five food supply chain models and the breadth of research along the three dimensions of sustainability (economic, environmental and social) represent the knowledge contribution that this study provides to the sustainable food supply chain research field.

From the practical implication perspective, food companies may find in this study a guide to better understand how to pursue the sustainability issues in their business, discovering useful information from the current research routes. For example, through this study, food companies could increase the awareness in the following themes: efficiency in food supply chain management; management of environmental, social and economic impacts deriving from the business activities; technologies adoption to foster agri-food sustainability; challenges and solutions in the agri-food industry; standards and regulations to foster sustainability. Moreover, food companies could increase the awareness on the possibility to implement sustainability along several dimensions following the different aspects discussed in environmental, social and economic sustainability. At last, also, the proposed food supply chain models linked to sustainability represent a driver to increase sustainability in food companies' activities.

### 6.2. Limitations

Despite that the research methodology was carefully defined according the PRISMA guidelines, some limits of this study can be debated. The research could be explained using other databases, other search schemes and following several kinds of analyses. Moreover, the results of the Science Mapping depend on the set parameters in VOSViewer. Choosing several values of minimal recurring term frequency and a different counting method, the starting network map could be different and could be composed by different clusters. Moreover, more or fewer terms could describe each cluster depending on the chosen term frequency threshold.

Regarding future developments, a further layer of knowledge can be added to the first one created in this study by analyzing the sample chosen using other kinds of analysis focused on contents, such as systematic content analysis or semantic analysis. These procedures allow researchers to apply selection criteria to the sample in order to refine the starting knowledge base and obtain more focused results or to analyze a different portion of the sample (e.g., book data source vs. journal data source) in order to compare the retrieved results in the sub-sample. Moreover, from this study, it emerges that sustainability is widely studied according the three dimensions but, refocusing the attention on specific themes, results are related to a specific dimension. For example, the food supply chain theme is addressed by an environmental viewpoint, the current agri-food challenges and strategies reflect the social and economic dimensions of sustainability. This opens new research routes encouraging academics and researcher to focus their efforts on investigating, equally, the three dimensions of sustainability. The overlay visualization of the thematic map in Figure 15, the timing evolution of the emerging current research routes in Figure 16 and the time visualization of sustainability related terms in Figure 17 provide to the researcher a way to evaluate how to change the direction of investigation in the oldest research route and how to continue the investigation in the youngest ones. For example, the investigation of food standards and regulations that resulted as mature could find new research routes in the analysis of the role that international agri-food certifications play in sustainable development. Similarly, the youngest research themes related to food insecurity and the impacts that pandemic risk generate in agri-food suggest that these research routes require more investigation in the future.

**Author Contributions:** Conceptualization, M.E.L.; methodology, M.M. and M.E.L.; software M.M.; validation, M.E.L.; formal analysis, M.M., M.D.G.; investigation, M.E.L. and M.M.; resources, M.E.L.; data curation, M.M., M.D.G.; writing—original draft preparation, M.E.L., M.M., M.D.G.; writing—

review and editing, M.E.L., M.M.; visualization, M.E.L., M.M.; supervision, M.E.L.; funding acquisition, M.E.L. All authors have read and agreed to the published version of the manuscript.

**Funding:** This research received no external funding.

**Institutional Review Board Statement:** Not applicable.

**Informed Consent Statement:** Not applicable.

**Data Availability Statement:** Data sharing not applicable.

**Acknowledgments:** This work was partially supported by REFIN intervention co-financed by the European Union through the Apulian regional plan (POR Puglia 2014–2020, Asse prioritario OT X "Investire nell'istruzione, nella formazione e nella formazione professionale per le competenze e l'apprendimento permanente"–Azione 10.4–DGR 1991/2018– Avviso2/FSE/2020 n. 57 del 13/05/2019–BURP n. 52 del 16/05/2019).

**Conflicts of Interest:** The authors declare no conflict of interest.

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
