# Peer review of "Evaluating the Sustainability Dimensions in the Food Supply Chain: Literature Review and Research Routes"

_sustainability, doi:10.3390/su132111816_

Round 1

Reviewer 1 Report

Overall an excellent manuscript. A few of my observations on the manuscript are as follows:

(i) This is more a kind of a procedural or methodological paper as a major portion of the manuscript is spent discussing research methods.

(ii) Further explaination on how you completed the research on a single day (September 3, 2021) would be helpful. Authors/researchers need to be thanked for this extraordinary work.

(iii) Were you able to find and review the entire paper or the whole write up of the 716 literature that you used for this research? What contents or parts of the literature did you use for analysis? Please clarify this in the method section. How have the validity and reliablity of the research been ascertained? 

(iv) Did you find any variation in results by data sources--books, journal aticles, or conference proceedings? Obviously, although journal articles are shorter in length but are peer reviewed and have powerful contents and may deserve more weightage than contents from other two sources.

(v) One of the reasons you were not be able to find many research done in or literature written in early 2000s' or prior to that era because there was not much digitalization then.

(vi) Many of the literature reviewed were from the U.S. and Europe which also seems to be obvious given that these continents had/have resources to undertake research and other works on food supply chain. 

(vii) On page 5, 6 and 7, Figure 2, 3 and 4 are not showing in the texts.

(viii) I would suggest the authors to include some recommendations and implications for scholars and stakeholders working in sustainability field.

Overall a good manuscript.

Author Response

Dear Reviewer, thank you for your suggestion. You can find in attachment a detailed answer to the provided comments. 

Reviewer 2 Report

I really enjoyed reviewing the article entitled Evaluating the sustainability dimensions in the food supply chain: literature review and research routes. I found this a very interesting article and one that can contribute a lot to the literature on sustainability.

The authors have carried out a very thorough bibliometric review covering all the necessary aspects of this type of work.

although certain formatting errors such as paragraphs not justified or with different spacing need to be corrected. There are also several missing references that give an error message.

Author Response

(The authors gave the same response as above.)

Reviewer 3 Report

The authors conduct a systematic literature review on the sustainability of the food supply chain. They identify six main research trends and five types of emergent food supply chains. They conclude that sustainability in the food supply chain is studied along its three dimensions but the focus on particular issues is generally related to a specific dimension.

The issue of sustainability in the food supply chain is topical and relevant. Methods are overall adequately described although some points could be clarified. Figures are attractive, informative and clearly presented but some could be further explained. The introduction and the presentation of results need to be better worked out. The conclusions could be further refined. Readability and style need to be improved too. See the comments below for more details.

Broad comments:

  • In my opinion, the introduction does not introduce properly the issue and does not put into perspective what follows. It does not succeed in explaining the problem addressed by the article. While the problem addressed (and the results) are very broad, the introduction focuses on a relatively small number of issues and gives insufficient emphasis to aspects which are important. There is no clear structure. For example, the authors speak practically interchangeably about food systems and food supply chain and a big emphasis is put in long and short food supply chains. Even though very related, food systems and the food supply chain do not have the same definition. Why do the authors focus on the food supply chain instead of food systems? Long and short food supply chains are not the only way in which sustainability and the food supply chain are related. I would suggest improving the introduction´s structure and targeting it somewhat better. I would also suggest trying to relate better the concepts of sustainability and food supply chain.
  • I would suggest using another term instead of models in “the main food supply chain models”. It looks as if the authors were referring to research models and methodologies.
  • Regarding the methodology section, I have a few specific questions. When the authors talk about geographic distribution, do they refer to the country of birth/residence/nationality of the authors or the country under study? Is it the standard to take into consideration only one scientific database? Do the nodes in the thematic analysis correspond to each of the article´s keywords? Does it measure the strength of the links or just the connections? I would like to know too whether both the descriptive analysis and the thematic analysis correspond to the bibliometric analysis. All these issues are kind of confusing as currently written in the article. I would suggest explaining better the methodology, especially these specific issues.
  • Regarding the results section, I also have a few concerns. Figures 2-6 have been titled twice. I would suggest deleting the titles inside the figures. Some of the arguments need to be strengthened (for example, the arguments used in lines 235-247 to justify the geographical distribution of the literature need to be better worked out). In Section 4, I would suggest explaining clearly how to interpret figures 6-13. Moreover, the results are presented too much like an endless list of terms. I would suggest focusing more on the interpretation of results. I also think the presentation of results needs to be better worked out.
  • I think the authors do not achieve the aim of “providing to food companies a framework of the sustainable paths that can be followed along the supply chain and to researchers gaps and future research routes to explore”, as stated in lines 86-88. I would suggest improving the conclusions to include suggestions for future research and insights on sustainable approaches for food companies. I also think the main research trends identified are overly broad. Could they be further refined?

Author Response

(The authors gave the same response as above.)

Round 2

Reviewer 3 Report

The paper has been improved. In my opinion, the comments have been addressed appropriately. However, I think readability still needs to be somewhat improved.